# Clinical benefit of cancer drugs approved in Switzerland 2010–2019

**Roman Adam**[1], **Ariadna Tibau**[2], **Consolación Molto Valiente**[2], **Boštjan Šeruga**[3], **Alberto Ocaña**[4], **Eitan Amir**[5], **Arnoud J. Templeton**[1,6,7]*

**1** Faculty of Medicine, University of Basel, Basel, Switzerland, **2** Oncology Department, Departament de Medicina de la Universitat Autònoma de Barcelona, Hospital de la Santa Creu i Sant Pau, Institut d'Investigació Biomèdica Sant Pau, Barcelona, Spain, **3** Institute of Oncology Ljubljana and Faculty of Medicine, Department of Medical Oncology, University of Ljubljana, Ljubljana, Slovenia, **4** Experimental Therapeutics Unit, Medical Oncology Department, Hospital Clínico Universitario San Carlos and IdISSC, Madrid, Spain, **5** Division of Medical Oncology and Hematology, Department of Medicine, Princess Margaret Cancer Center and the University of Toronto, Toronto, ON, Canada, **6** Department of Medical Oncology, St. Claraspital, Basel, Switzerland, **7** St. Clara Research Ltd., Basel, Switzerland

* arnoud.templeton@claraspital.ch

**Data Availability Statement:** All relevant data are within the paper and its Supporting Information files.

**Funding:** The authors received no specific funding for this work.

## Abstract

### Background

It is unknown to what extent cancer drugs approved in Switzerland by the Swissmedic fulfil criteria of clinical benefit according to the European Society of Medical Oncology Magnitude of Clinical Benefit Scale version 1.1 (ESMO-MCBS), the American Society of Clinical Oncology Value Framework version 2 (ASCO-VF) and the Swiss OLUtool v2 (OLUtool).

### Patients and methods

An electronic search identified studies that led to marketing authorisations in Switzerland 2010–2019. Studies were evaluated according to ESMO-MCBS, ASCO-VF and OLUtool. Substantial benefit for ESMO-MCBS, was defined as a grade A or B for (neo)adjuvant intent and 4 or 5 for palliative intent. For ASCO-VF and OLUtool clinical benefit was defined as score ≥45 and A or B, respectively. Concordance between the frameworks was calculated with Cohen's Kappa (κ). Factors associated with clinical benefit were evaluated by logistic regression.

### Results

In the study period, 48 drugs were approved for 92 evaluable indications, based on 100 studies. Ratings for ESMO-MCBS, ASCO-VF and OLUtool could be performed for 100, 86, and 97 studies, respectively. Overall, 39 (39%), 44 (51%), 45 (46%) of the studies showed substantial clinical benefit according to ESMO-MCBS v1.1, ASCO-VF, OLUtool criteria, respectively. There was fair concordance between ESMO-MCBS and ASCO-VF in the palliative setting (κ = 0.31, $P$ = 0.004) and moderate concordance between ESMO-MCBS and OLUtool (κ = 0.41, $P$<0.001). There was no significant concordance between ASCO-VF and OLUtool (κ = 0.18, $P$ = 0.12). Factors associated with substantial clinical benefit in multivariable analysis were HRQoL benefit reported as secondary outcome for ESMO-MCBS and the ASCO-VF and blinded studies for OLUtool.

**Competing interests:** I have read the journal's policy and the authors of this manuscript have declared the following competing interests (and leave it to the judgement of the editor whether this is of relevant for the submitted work): Eitan Amir: reports personal fees for expert testimony from Genentech/Roche and an advisory role for Sandoz, Novartis and Exact Sciences. Ariadna Tibau: reports personal fee for travel grant from Pfizer and honoraria from Eisai, Roche and Novartis outside the submitted work. Arnoud J. Templeton: advisory board/consultancy: Astellas, MSD, BMS (institution), Janssen (institution), Sanofi (institution), Roche (institution); honoraria: Astellas, Sanofi; conference/travel support: Bayer, Sanofi, Janssen, Ipsen, Roche. This does not alter our adherence to PLOS ONE policies on sharing data and materials. All remaining authors have declared no conflict of interest.

## Conclusions

At the time of approval, only around half of the trials supporting marketing authorisation of recently approved cancer drugs in Switzerland meet the criteria for substantial clinical benefit when evaluated with ESMO-MCBS, ASCO-VF or OLUtool. There was at best only moderate concordance between the grading systems.

## Background

Over time, the number of cancer drugs approved by the Food and Drug Administration (FDA) has increased [1]. This has been attributed to advances in cancer drug research but also to faster approvals which are based often on preliminary data, and intermediate primary endpoints instead of overall survival (OS) [2,3]. While it is desirable to make treatments available on the market in a shorter time, this should not happen at the expense of effectiveness or safety of the approved substances and must therefore be approached with caution. There are concerns that faster drug approval may be associated with lower drug efficacy and worse patient safety [4,5]. Possibly also partly due to the high amount of approvals based on intermediate endpoints [6,7] whose correlation to long-term outcomes like OS and Health related quality of life (HRQoL) was low in studies [8–10].

Effective treatments should prolong survival time and/or improve HRQoL. Previous studies have argued that certain cancer drugs approved by the FDA [11] and the European Medicines Agency (EMA) [12] show questionable clinical benefit. In Switzerland, approval for new drugs is granted by the Swiss national authorisation and supervisory authority Swissmedic. In the authorisation process of medical products Swissmedic takes into account pharmacological and clinical data on efficacy and safety as well as HRQoL. The price of a medicinal product is not assessed by Swissmedic and is determined after authorisation either by the authorisation holder or, in the case of an application for health insurance coverage, by the Swiss Federal Office of Public Health.

There are different validated tools to assess the clinical benefit of a cancer drugs: Commonly used tools are the European Society for Medical Oncology–Magnitude of Clinical Benefit Scale version 1.1 (ESMO-MCBS v1.1) [13], the American Society of Clinical Oncology—Value Framework version 2 (ASCO VF v2) [14], and in Switzerland the OLUtool Onko version 2.0 (OLUtool v2) [15]. These tools were developed with different goals. The ESMO-MCBS was created to quantify the effectiveness of new cancer treatments [16], the ASCO-VF was developed to facilitate physicians and patients to assess the expected clinical benefit of a cancer treatment to help them in their decision making process [17], and the OLUtool is used for the case-by-case decision on the reimbursement of treatment costs in the off-label use of cancer drugs [15].

Creating a validated tool to quantify the clinical benefit of cancer treatments is complex and takes into account different aspects on the efficacy, toxicity and HRQoL. The different selection and weighting of these data can lead to differences in the results of the application of the frameworks. Previous studies that assessed the concordance between the ESMO-MCBS and the ASCO-VF have shown varying results ranging from fair [18] to substantial [19] concordance.

The aim of our study was to investigate to what extent the cancer drugs approved in Switzerland between January 1 2010 and December 31 2019 fulfil the criteria for a substantial clinical benefit when evaluated with the ESMO-MCBS v1.1, the ASCO-VF v2 and the OLUtool v2. Furthermore, we investigated whether there was concordance between the grading tools and evaluated factors associated with clinical benefit.

## Methods

### Data sources, study selection and data extraction

Drugs used for treatment of solid tumours which received first marketing authorisation between January 2010 and December 2019 were identified from the official Swissmedic Journals and the website swissmedicinfo.ch [20]. Subsequently, we searched for pivotal studies supporting the approved drug. Next, studies which reported an endpoint evaluable by ESMO-MCBS v1.1, ASCO-VF v2, or OLUtool Onko v2, namely overall survival (OS), progression-free survival (PFS), disease-free survival (DFS), relapse-free survival (RFS), and objective response rate (ORR) were identified. Updated data of pivotal studies or reports on toxicity or QoL outcomes were considered in the analysis only if they were published before the date of Swiss marketing approval of a respective drug for a given indication.

Drug names, indications and approval dates were recorded from the official website and the following data were subsequently extracted from publications: name of first author, publication date, journal, publication type (abstract / full paper), primary cancer site, number of patients evaluated, study design (randomized vs. single-arm), blinding (double-blind vs. open-label), phase of study (phase I, II, III), drug class, treatment setting (curative vs. palliative), primary and secondary study endpoints (OS, PFS, DFS, RFS, ORR), data on toxicity and on HRQoL, cross-over (yes vs. no), subgroup analysis (yes vs. no), survival curve plateau (yes vs. no) and the need for companion diagnostics.

If the indication corresponded to a specific subgroup evaluated in a study, the data for the subgroup were evaluated if they were reported separately in the study. If the indication was based on different studies with different selection criteria (e.g. mutation status, prior therapies) or studies of different treatment lines, all these studies were considered. If the indication was expanded to include other subgroups (e.g. addition of a further mutation class or new line or treatment, e.g. additional approval as first-line) the study data supporting the expansion were evaluated. In cases where different studies with identical selection criteria supported approval, the study with the biggest sample size was selected. If different subgroups were reported in a trial, we selected the subgroup which best matched the indication.

### Data synthesis, scoring

All identified studies were evaluated using the ESMO-MCBS v1.1, the ASCO-VF v2 and OLUtool v2 according to published guidelines [13,14,21]. For the grading of the studies, only statistically significant data with a p-value $< 0.05$ were considered. Additionally, HRQoL outcomes had to be based on a validated questionnaire and show a clinically relevant improvement. Substantial clinical benefit was defined as 5 or 4 and A or B in the ESMO-MCBS in the palliative and curative setting, respectively. For ASCO-VF v2 a conservative threshold for substantial benefit of 45 points was defined as previously reported by Cherny et al. [19].

We performed a sensitivity analysis for agreement in which we only compared studies that were evaluated for the same efficacy endpoints. To evaluate if the previously reported threshold of 45 points for a substantial clinical benefit for the ASCO-VF v2 also fits our data we generated a receiver-operating characteristic (ROC) curve to find the optimal threshold for substantial clinical benefit for our data and calculated the concordances in this analysis.

### Statistical analysis

Study data are reported as median, ranges and point estimates of time to event endpoints, where appropriate. The agreement between the different frameworks was calculated using

Cohen's Kappa (for dichotomous outcomes) and Spearman's ranks correlation (for continuous outcomes). Interpretation of correlations and concordance was performed according to previous studies [22,23]. Trends over time were calculated by linear regression and their p-values were reported. We used box plots to show the ratings of the ASCO-VF v2 scores in relation to the ESMO-MCBS v.1.1 scores and the OLUtool v2 grades visually. Independent predictors (shown in S4 Table) were calculated using univariable and multivariable logistic regression and stated as Odds ratio (OR) with corresponding 95% confidence intervals (CI) and p-values. Only variables with a p-value < 0.1 in the univariable analysis were included in multivariable analyses. SPSS version 25 (IBM Corp, Armonk NY) was used for all analyses. All statistical tests were two sided and significance level was defined as p-value < 0.05. No correction for multiple statistical testing was applied.

## Results

### Drugs approved

During the study period from 2010 to 2019, Swissmedic approved 48 new cancer drugs for 101 indications in solid tumours. Overall, 100 studies supporting the approvals of 92 indications of 45 new cancer drugs were found (Fig 1). Study characteristics are presented in Table 1. Most studies were in the palliative setting and most of them were randomized phase 3 trials. Although the HRQoL was evaluated in around half of the studies it was only reported and thus evaluable as a secondary outcome in around one third of the studies.

### Substantial clinical benefit

Evaluations with ESMO-MCBS v.1.1, ASCO-VF v2 and OLUtool v2 could be performed for 92, 78, and 90 indications supported by 100, 86, and 97 studies, respectively. At the time of approval, 39 (39%), 44 (51%), and 45 (46%) of all evaluated studies met the criteria for substantial clinical benefit according to ESMO-MCBS v1.1, ASCO-VF v2 and OLUtool v2, respectively (Fig 2 and S1 Table).

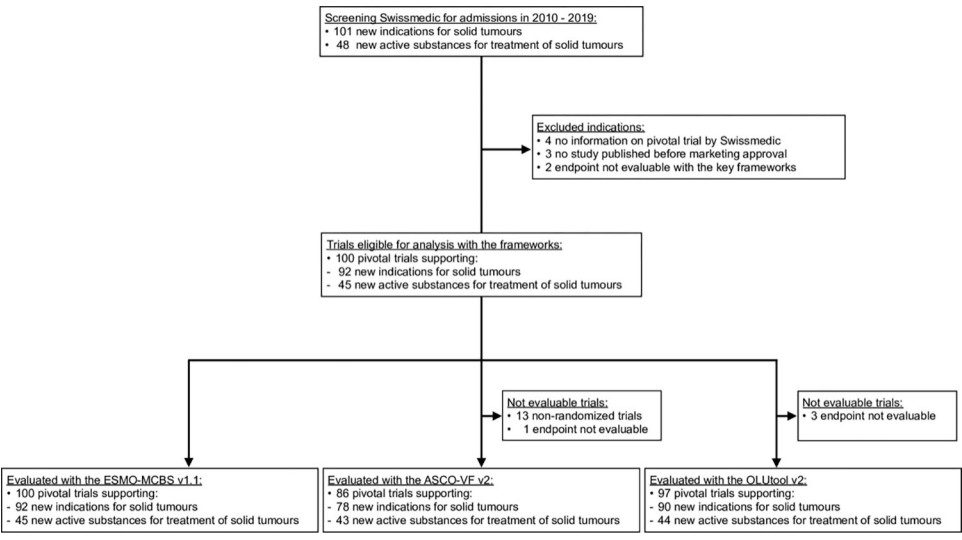

**Fig 1. Identification of new active substances, indications for solid tumours and their supporting pivotal trials published between January 2010 and December 2019 which showed an endpoint evaluable with the ESMO-MCBS v1.1, the ASCO-VF v2 and the OLUtool v2.**

**Table 1. Characteristics of pivotal trials evaluable with the ESMO-MCBS v1.1, the ASCO-VF v2 and the OLUtool v2.**

| Framework | | ESMO-MCBS v1.1 | ASCO-VF v2 | OLUtool v2 |
|---|---|---|---|---|
| Characteristics | | N (%) | N (%) | N (%) |
| Studies available | | 100 (100) | 86 (100) | 97 (100) |
| Randomized Patients (patients included in the study) | | | | |
| | Median | 548 | 599 | 553 |
| | Range | 50–4805 | 133–4805 | 50–4805 |
| Analysed Patients | | | | |
| | Median | 444 | 556 | 493 |
| | Range | 50–4804 | 79–4804 | 50–4804 |
| Tumour type | | | | |
| | Lung cancer | 23 (23) | 17 (20) | 23 (24) |
| | Melanoma | 17 (17) | 16 (19) | 15 (16) |
| | Breast cancer | 17 (17) | 15 (17) | 16 (17) |
| | Gastrointestinal (including hepatocellular) | 10 (10) | 8 (9) | 10 (10) |
| | Prostate cancer | 7 (7) | 7 (8) | 7 (7) |
| | Ovarian cancer | 6 (6) | 6 (7) | 6 (6) |
| | Sarcoma (including GIST) | 5 (5) | 5 (6) | 5 (5) |
| | Renal cancer | 5 (5) | 5 (6) | 5 (5) |
| | Urothelial | 3 (3) | 2 (2) | 3 (3) |
| | Other | 7 (7) | 5 (6) | 7 (7) |
| Setting | | | | |
| | Curative (neoadjuvant/adjuvant) | 7 (7) | 6 (7) | 6 (6) |
| | Palliative | 93 (93) | 80 (93) | 91 (94) |
| Line of treatment | | | | |
| | Neoadjuvant/Adjuvant | 7 (7) | 6 (7) | 6 (6) |
| | First line | 43 (43) | 38 (44) | 42 (43) |
| | Second line | 43 (43) | 35 (41) | 42 (43) |
| | Third line | 7 (7) | 7 (8) | 7 (7) |
| Study design | | | | |
| | Randomized | 87 (87) | 86 (100) | 84 (87) |
| | Single-arm | 13 (13) | 0 | 13 (13) |
| Phase of study | | | | |
| | Phase 1 | 2 (2) | 0 | 2 (2) |
| | Phase 2 | 17 (17) | 5 (6) | 16 (17) |
| | Phase 3 | 81 (81) | 81 (94) | 79 (81) |
| Blinding | | | | |
| | Open-label | 51 (51) | 37 (43) | 48 (49) |
| | Double-blind | 49 (49) | 49 (57) | 49 (51) |
| Crossover | | | | |
| | Allowed | 19 (25) | 19 (26) | 19 (26) |
| | Not allowed | 56 (75) | 55 (74) | 54 (74) |
| Time-to-event as primary endpoint | | | | |
| | Yes | 84 (84) | 84 (98) | 83 (86) |
| | No | 16 (16) | 2 (2) | 14 (14) |
| Number of primary endpoints | | | | |
| | 1 | 80 (80) | 67 (78) | 78 (80) |
| | >1 | 20 (20) | 19 (22) | 19 (20) |
| Primary endpoint | | | | |

*(Continued)*

**Table 1.** (Continued)

| Framework | | ESMO-MCBS v1.1 | ASCO-VF v2 | OLUtool v2 |
|---|---|---|---|---|
| | Overall survival | 41 (41) | 41 (48) | 40 (41) |
| | Progression or disease free survival | 54 (54) | 54 (63) | 54 (56) |
| | Objective response rate | 16 (16) | 3 (4) | 15 (16) |
| Health related quality of life reported | | | | |
| | Yes | 52 (52) | 48 (56) | 52 (54) |
| | No | 48 (48) | 38 (44) | 45 (46) |
| Health related quality of life reported as secondary outcome | | | | |
| | Yes | 32 (32) | 32 (37) | 32 (33) |
| | No | 68 (68) | 54 (63) | 65 (67) |
| Improvement in HRQoL reported as secondary outcome | | | | |
| | Yes | 15 (15) | 15 (17) | 15 (15) |
| | No | 85 (85) | 71 (83) | 82 (85) |
| Companion diagnostics | | | | |
| | Yes | 41 (41) | 30 (35) | 40 (41) |
| | No | 59 (59) | 56 (65) | 57 (59) |
| Experimental drug | | | | |
| | Small molecule (TKI, PARPi, etc.) | 37 (37) | 29 (34) | 37 (38) |
| | Immune checkpoint inhibitor monotherapy | 22 (22) | 19 (22) | 21 (22) |
| | CDK4/6 inhibitor plus endocrine therapy | 7 (7) | 7 (8) | 7 (7) |
| | Immune checkpoint inhibitor combination | 7 (7) | 6 (7) | 7 (7) |
| | Endocrine therapy | 6 (6) | 6 (7) | 6 (6) |
| | Non-checkpoint inhibitor antibody combination | 6 (6) | 5 (6) | 5 (5) |
| | Small molecule combination | 5 (5) | 4 (5) | 5 (5) |
| | Chemotherapy monotherapy | 4 (4) | 4 (5) | 4 (4) |
| | Other | 6 (6) | 6(7) | 5 (5) |

Abbreviations: N: number; GIST: Gastrointestinal stromal tumour; HRQoL: Health related Quality of life; TKI: Tyrosine kinase inhibitor; PARPi: Poly (ADP-ribose) polymerase inhibitor; TKI: Tyrosine kinase inhibitor; PARPi: Poly (ADP-ribose) polymerase inhibitor; CDK4/6: Cycline dependent kinase 4/6.

## Concordance and correlation of grading systems

The number of concordant studies, concordances and correlations between the grades of the studies evaluated with the different frameworks are presented in Table 2. Overall, there was

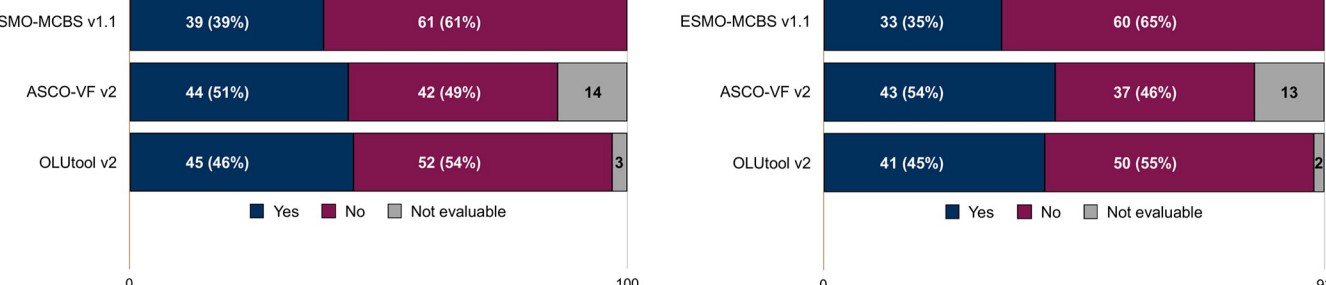

**Fig 2. Number of studies which fulfilled the criteria for substantial clinical benefit for the respective frameworks.** A. All studies; B: studies in the palliative setting.

**Table 2. Concordance and correlation between the different frameworks.**

| | ESMO-MCBS v1.1/ASCO-VF v2 | | | ESMO-MCBS v1.1/OLUtool v2 | | | ASCO-VF v2/OLUtool v2 | | |
|---|---|---|---|---|---|---|---|---|---|
| | all studies (N = 86) | palliative setting (N = 80) | curative setting (N = 6) | all studies (N = 97) | palliative setting (N = 91) | curative setting (N = 6) | all studies (N = 84) | palliative setting (N = 78) | curative setting (N = 6) |
| Number of concordant studies | 53 (62%) | 52 (65%) | 1 (17%) | 69 (71%) | 65 (71%) | 4 (67%) | 49 (58%) | 46 (59%) | 3 (50%) |
| Spearman's rho | | 0.42 ($P<0.001$) | | | 0.58 ($P<0.001$) | | | 0.40 ($P<0.001$) | |
| Cohen's Kappa | 0.23 ($P = 0.029$) | 0.31 ($P = 0.004$) | | 0.41 ($P<0.001$) | 0.41 ($P<0.001$) | | 0.16 ($P = 0.133$) | 0.18 ($P = 0.121$) | |

Abbreviations: ESMO-MCBS v1.1: European Society for Medical Oncology—Magnitude of Clinical Benefit Scale version 1.1; ASCO-VF v2: American Society of Clinical Oncology—Value Framework version 2; OLUtool v2: OLUtool version 2; *P*: p-value.

fair to moderate concordance between the different frameworks and was highest for the comparison of ESMO-MCBS v1.1 with the OLUtool v2 (kappa 0.41, p < 0.001). Higher scores in the ASCO-VF v2 were associated with higher grades from ESMO-MCBS v1.1 and OLUtool v2 (Fig 3) and showed fair correlation between the tools (Table 2). ROC curve analyses suggested that 47 points and 56 points has the highest discriminatory ability in the studies analysed for the comparison with the ESMO-MCBS v1.1 and the OLUtool v2, respectively (S2 Table). Considering studies in the palliative setting only (N = 93) the optimal cut-offs were 49 and 56, respectively.

In sensitivity analyses concordance was similar when comparing the overall study selection and the palliative setting and when considering studies with the same endpoints only (S3 Table).

## Trends over time

The proportion of studies supporting drug approval in Switzerland with results indicating substantial clinical benefit has largely remained unchanged over the study period (Fig 4). Also, when only considering studies in the palliative setting there was no statistically significant trend over time for the rate of studies with substantial clinical benefit evaluated with the ESMO-MCBS ($P_{trend}$ = 0.13) or the ASCO-VF ($P_{trend}$ = 0.72). Of note, for OLUtool v2 the number of trials meeting substantial clinical benefit significantly increased over time (2010–2012: 31%, vs. 2017–2019: 53%; $P_{trend}$ = 0.032) suggesting that more recently approved indications more frequently fulfil the criteria for a substantial clinical benefit.

## Predictors for substantial clinical benefit

In univariable analyses the benefit of HRQoL as secondary outcome was associated with a higher likelihood of substantial clinical benefit for ESMO-MCBS v1.1 and ASCO-VF v2 (S4 Table). In the ESMO-MCBS v1.1 and OLUtool v2 phase 3 studies had a higher likelihood for a substantial clinical benefit while this was not the case with ASCO VF v2. Studies for treatments with a curative intent and studies in which crossover was allowed were also predictive of a substantial clinical benefit when evaluated with the ESMO-MCBS v1.1 and for the OLUtool blinded studies showed higher odds for a substantial clinical benefit. The association between HRQoL benefit for the ESMO-MCBS v1.1 and ASCO-VF v2 and blinded studies for the OLUtool v2 mentioned above were maintained in the multivariable analysis (Table 3). Other associations were no longer significant.

**Distribution of the scores evaluated with the ASCO-VF v2 and the ESMO-MCBS v1.1**

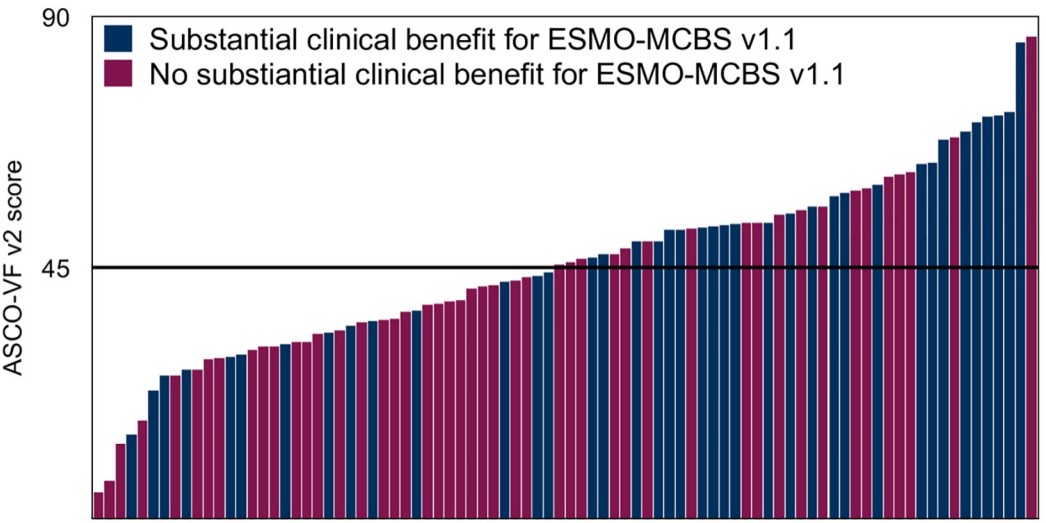

**Distribution of the scores evaluated with the ASCO-VF v2 and the OLUtool v2**

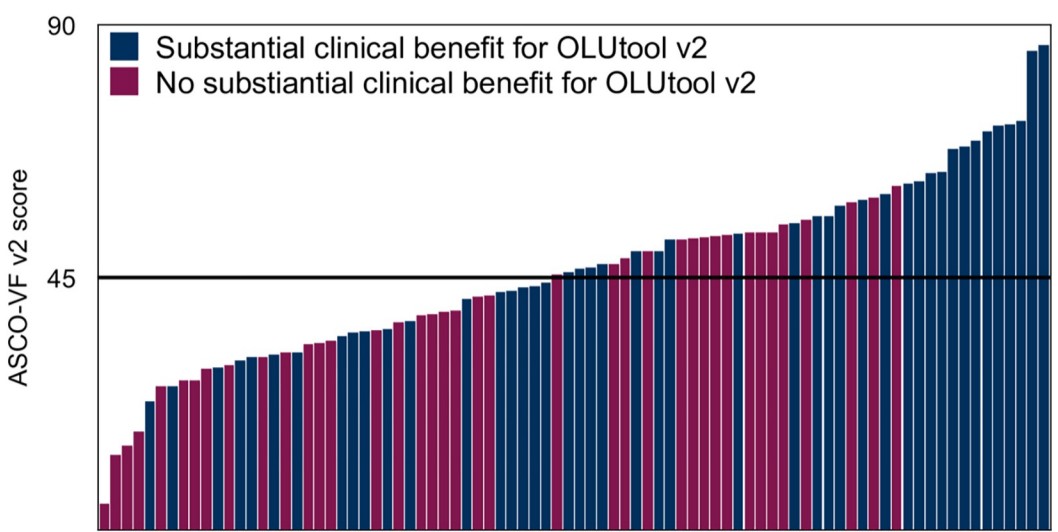

**Boxplots of the grades evaluated with the ESMO-MCBS v1.1, the ASCO-VF v2 and the OLUtool v2**

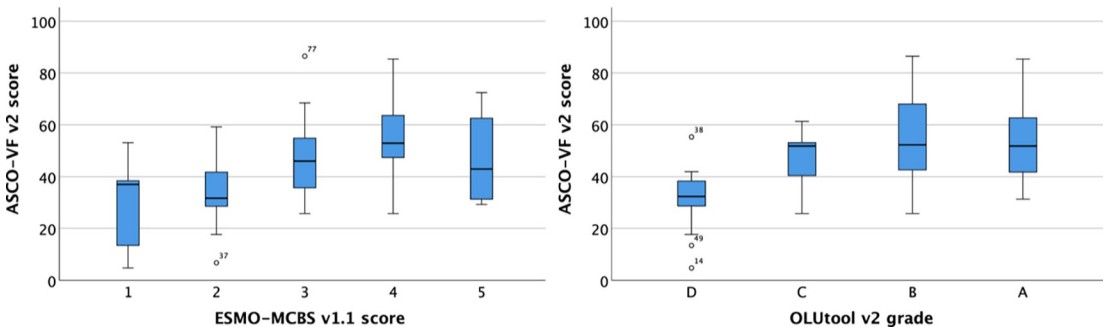

**Fig 3. Scores evaluated with the ASCO-VF-v2 according to their ESMO-MCBS v1.1 scores (A/C) or OLUtool v2 grades (B/C).**

## Discussion

We evaluated the pivotal trials supporting approval of cancer drugs in Switzerland during the last decade using the ESMO-MCBS v1.1, ASCO-VF v2 and the OLUtool v2 to evaluate clinical benefit according to these well-characterized frameworks. We found that only around half of the included trials showed a substantial clinical benefit at the time of approval. These findings are in line with analyses of studies supporting cancer drug approvals by the FDA, the EMA and the Canadian Agency for Drugs and Technologies in Health [11,12,24]. This result also suggests that the Swiss regulatory body Swissmedic evaluates the effectiveness of the treatments differently than the frameworks examined in our study, likely emphasizing formal statistical positivity of studies irrespective of the clinical impact of the primary endpoint.

We also evaluated the concordance of the different frameworks and found discrepancies in the grades which likely originate in the different methodical approaches and selection of the analysed data as discussed by Cherny and colleagues [19]. One of the major differences between the frameworks is the use of the point estimate of the hazard ratio rather than the lower-limit of the 95% confidence interval in the ASCO-VF v2 and the OLUtool v2, as compared to the ESMO-MCBS v1.1, respectively. Second, the frameworks also differ in the way how the toxicity of the treatments is considered. While the ESMO-MCBS v1.1 and the OLUtool v2 consider the absolute proportion of patients with high-grade adverse events in their ratings, the ASCO-VF v2 bases its toxicity adjustment on the relative occurrence of adverse events of all grades. In addition, in the OLUtool v2 and the ESMO-MCBS v1.1, only one bonus point is possible for adjustment, whereas for the ASCO-VF v2 bonus points are often awarded for reduction of toxicity and for improvement of HRQoL, factors that often go hand in hand in a clinical trial. The frameworks also differ in how they credit the tail of the curve and it seems more difficult to obtain a bonus for a tail of the curve with ESMO-MCBS v1.1 and OLUtool v2 than with the ASCO-VF v2. In general, this means that outcomes that are used for adjustments (such as toxicity and HRQoL) might have a cumulative effect with ASCO-VF v2 in the non-curative setting whereas in ESMO-MCBS v1.1 and the OLUtool v2 this is not the case.

Differences between the ratings are likely to also have resulted from studies utilizing different efficacy endpoints. Additionally, the threshold of 45 or greater points for ASCO-VF v2 may not have been optimal for our study selection. This is shown by the sensitivity analyses, where studies for which the same endpoint was evaluated in both compared frameworks and for which the optimal threshold was calculated with the ROC curve showed higher concordance. Indeed, when using the established thresholds of grade A or B with OLUtool v2 the optimal cut for substantial clinical benefit when using the ASCO-VF v2 was 56 points. However, this cut-off resulted from the comparison with OLUtool v2 which was originally created to help with the decision of cost coverage in the off-label setting and thus uses more rigorous criteria than can be expected for registration trials, likely with a higher threshold.

Apart from the grades for the OLUtool v2, there was no significant changes in scores over time. However, there is an encouraging trend that more recent registration trials evaluate HRQoL more often as a secondary outcome than this was done in older registration trials. Some of these data on HRQoL were not yet published at the time of approval and were therefore not included in our analysis. However, considering that HRQoL was an independent predictor for substantial clinical benefit in ESMO-MCBS v1.1 and ASCO-VF v2, it is likely that the ratings of the pivotal trials would be higher when including these HRQoL data.

**Substantial clinical benefit per year: ESMO-MCBS v1.1**

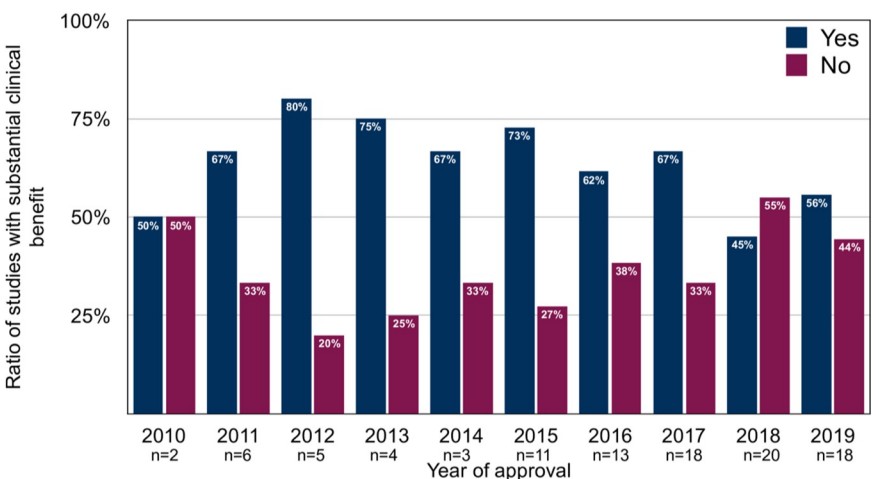

**Substantial clinical benefit per year: ASCO-VF v2**

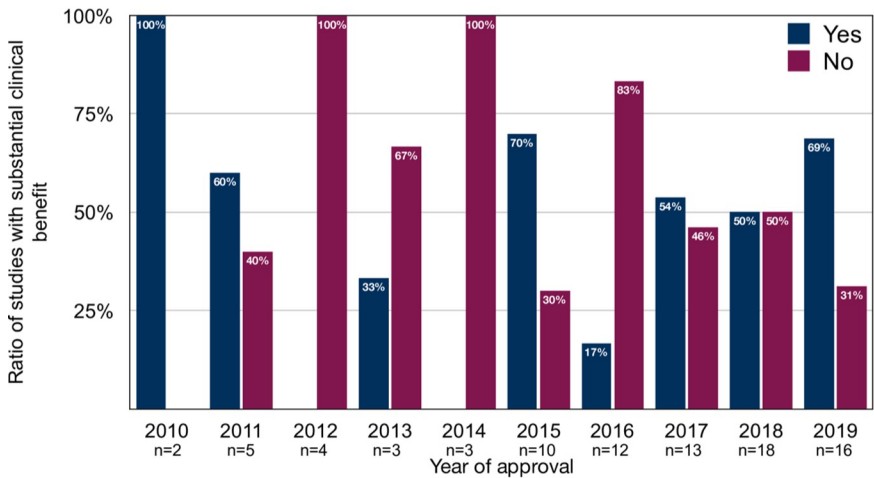

**Substantial clinical benefit per year: OLUtool v2**

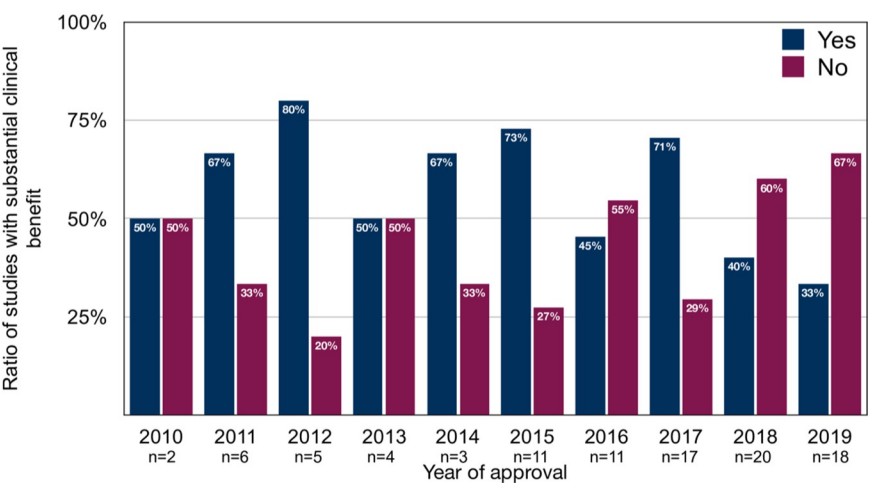

**Fig 4. Proportion of of studies meeting the criteria for a substantial clinical benefit from 2010 to 2019. A**: ESMO-MCBS 1.1; **B**: ASCO-VF v2; **C**: OLUtool v2.

**Table 3. Predictors for substantial clinical benefit on multivariable logistic regression for all studies.**

| | | | ESMO-MCBS v1.1 (all studies) | | ASCO-VF v2 (all studies) | | OLUtool v2 (all studies) | |
|---|---|---|---|---|---|---|---|---|
| **Multivariable analysis** | | | OR (95% CI) | *P* | OR (95% CI) | *P* | OR (95% CI) | *P* |
| | Phase 3 (vs. phase 1, 2) | | 4.65 (2.03–124.86) | 0.235 | | | 3.47 (0.86–13.98) | 0.080 |
| | Cross-over allowed (versus not) | | 3.23 (1.05–9.09) | 0.056 | | | | |
| | Line of treatment | | | | | | | |
| | | Neoadjuvant/adjuvant | 8.15 (1.26–94.52) | 0.142 | | | | |
| | | First line (vs. further line) | | | | | 2.06 (0.83–5.12) | 0.121 |
| | Blinded study (vs. open label) | | | | 0.69 (0.27–1.80) | 0.451 | **3.30 (1.30–8.40)** | **0.012** |
| | HRQoL benefit as secondary outcome (yes vs. no) | | **6.33 (2.24–32.98)** | **0.012** | **4.35 (1.04–18.15)** | **0.044** | | |
| | Approved since 2017 (vs. 2010–2016) | | | | 0.47 (0.19–1.16) | 0.100 | | |

Abbreviations: ESMO-MCBS v1.1: European Society for Medical Oncology—Magnitude of Clinical Benefit Scale Version 1.1; ASCO-VF v2: American Society of Clinical Oncology—Value Framework Version 2; OLUtool v2: OLUtool Version 2; OR: odds ratio; 95% CI: 95% confidence interval; P: p-value; vs.: versus; incl.: inclusive; HRQoL: Health related quality of life. Only variables with a p-value < 0.1 in the univariable analysis (for details see S4 Table) were included in multivariable analyses.

Our study has several limitations. First, we limited our analysis to solid tumours and excluded drugs approved to treat hematologic malignancies. Second, Swissmedic does not provide the exact data on which the marketing approval is granted and since they constantly update the study data on their webpage, the data do not always correspond to the data available at the time of authorisation. In order to address this problem, we decided to search for the original data of the pivotal trials and their updates and evaluate only data that had already been published before the date of the Swiss marketing authorisation. However, it is possible that Swissmedic had access to updated data which were not published at the time of approval and thus could not be included in our analysis. Third, our data were retrieved from published articles only. Therefore, data on HRQoL were missing frequently and data on toxicity was often based on pooled analyses. Due to the different methods of considering data on toxicity, adjustments made for toxicity in the different frameworks can differ depending on the way data is presented by the authors. Fourth, although OLUtool v2 was developed to evaluate off-label treatments, in this study we used it to rate on-label treatments where higher level evidence might be expected.

In summary, at the time of approval, around half of the pivotal studies supporting cancer drug approvals during the last decade in Switzerland meet criteria for substantial clinical benefit as rated with various frameworks (ASCO-VF v2, ESMO-MCBS v1.1, OLUtool v2). These frameworks only have fair to moderate concordance suggesting different appreciation of endpoints and magnitude of effect reported in studies. Further research is needed to establish optimal rating systems to determine and compare clinical benefits of new cancer drugs in clinical studies and in daily practice.

## Supporting information

**S1 Table. Grades evaluated with the ESMO-MCBS v1.1, the ASCO-VF v2 and the OLUtool v2.**
(DOCX)

**S2 Table. Sensitivity analysis for the correlation and concordance between the studies with the optimal threshold calculated with ROC curve analyses for ASCO-VF v2.**
(DOCX)

**S3 Table. Sensitivity analysis for the concordance and the correlation between the studies considering studies with the same evaluated endpoints only.**
(DOCX)

**S4 Table. Predictors of substantial clinical benefit on univariable logistic regression for all studies.**
(DOCX)

**S1 File.**
(XLSX)

## Author Contributions

**Conceptualization:** Ariadna Tibau, Arnoud J. Templeton.

**Data curation:** Roman Adam, Consolación Molto Valiente, Arnoud J. Templeton.

**Formal analysis:** Roman Adam, Ariadna Tibau, Consolación Molto Valiente, Boštjan Šeruga, Alberto Ocaña, Eitan Amir, Arnoud J. Templeton.

**Investigation:** Roman Adam, Eitan Amir, Arnoud J. Templeton.

**Methodology:** Roman Adam, Ariadna Tibau, Eitan Amir, Arnoud J. Templeton.

**Project administration:** Arnoud J. Templeton.

**Supervision:** Arnoud J. Templeton.

**Validation:** Ariadna Tibau, Consolación Molto Valiente, Boštjan Šeruga, Alberto Ocaña, Eitan Amir, Arnoud J. Templeton.

**Visualization:** Roman Adam, Arnoud J. Templeton.

**Writing – original draft:** Roman Adam, Arnoud J. Templeton.

**Writing – review & editing:** Ariadna Tibau, Consolación Molto Valiente, Boštjan Šeruga, Alberto Ocaña, Eitan Amir, Arnoud J. Templeton.

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
