## [Decision Letter · Decision Letter 0]

18 Mar 2022

PONE-D-21-28268Clinical Benefit of cancer drugs approved in Switzerland 2010 - 2019PLOS ONE

Dear Dr. Arnoud J Templeton,

Thank you for submitting your manuscript to PLOS ONE. After careful consideration, we feel that it has merit but does not fully meet PLOS ONE’s publication criteria as it currently stands. Therefore, we invite you to submit a revised version of the manuscript that addresses the points raised during the review process.

There are several concerns about the discuss section. Discuss with more evidences or references would improve the MS quality. Please submit your revised manuscript by April 30, 2022. If you will need more time than this to complete your revisions, please reply to this message or contact the journal office at plosone@plos.org. Please include the following items when submitting your revised manuscript:A rebuttal letter that responds to each point raised by the academic editor and reviewer(s). You should upload this letter as a separate file labeled 'Response to Reviewers'.A marked-up copy of your manuscript that highlights changes made to the original version. You should upload this as a separate file labeled 'Revised Manuscript with Track Changes'.An unmarked version of your revised paper without tracked changes. You should upload this as a separate file labeled 'Manuscript'.

We look forward to receiving your revised manuscript.

Kind regards,

Wen-Wei Sung, M.D., Ph.D.

Academic Editor

PLOS ONE

“I have read the journal's policy and the authors of this manuscript have declared the following competing interests (and leave it to the judgement of the editor whether this is of relevant for the submitted work):

Eitan Amir: reports personal fees for expert testimony from Genentech/Roche and an advisory role for Sandoz, Novartis and Exact Sciences.

Ariadna Tibau: reports personal fee for travel grant from Pfizer and honoraria from Eisai, Roche and Novartis outside the submitted work.

Arnoud J. Templeton: advisory board/consultancy: Astellas, MSD, BMS (institution), Janssen (institution), Sanofi (institution), Roche (institution); honoraria: Astellas, Sanofi; conference/travel support: Bayer, Sanofi, Janssen, Ipsen, Roche.

All remaining authors have declared no conflict of interest.”

3. We noted in your submission details that a portion of your manuscript may have been presented or published elsewhere. [DETAILS AS NEEDED] Please clarify whether this [conference proceeding or publication] was peer-reviewed and formally published. If this work was previously peer-reviewed and published, in the cover letter please provide the reason that this work does not constitute dual publication and should be included in the current manuscript.

Reviewers' comments:

Reviewer's Responses to Questions

**Comments to the Author**

1. Is the manuscript technically sound, and do the data support the conclusions?

Reviewer #1: Yes

Reviewer #2: Yes

2. Has the statistical analysis been performed appropriately and rigorously? 

Reviewer #1: Yes

Reviewer #2: Yes

3. Have the authors made all data underlying the findings in their manuscript fully available?

Reviewer #1: Yes

Reviewer #2: Yes

4. Is the manuscript presented in an intelligible fashion and written in standard English?

Reviewer #1: Yes

Reviewer #2: Yes

5. Review Comments to the Author

Reviewer #1: The authors of the mentioned manuscript investigated the clinical benefit of different solid cancer drugs approved in Switzerland as measured by different published scoring systems. The article discusses a relevant topic and is written in an intelligible fashion, the statistics are performed correctly and the results are discussed in a differentiated manner.

Reviewer #2: The authors collected over 100 studies of approved cancer drugs

and compared 3 different published frameworks that

measure the clinical benefit and efficacy of the new treatments.

Based on the measures of the frameworks only half of the approved drugs showed

a substantial clinical benefit. The study also represents an impressive collection

of drug efficacy and outcome measures and is

a very valuable contribution.

1) The study performs a global assessment of all studies and subsets and

would benefit from including some more details which would dramatically improve

the reading and understanding of the paper.

For example the data collection is unclear,

a list/table of the data matrix of those 100 drugs with the described endpoints and

the inputs that were used for each of the 3 frameworks, e.g.

OS, PFS, DFS, RFS, ORR, toxicity, QoL outcomes and missing data and framework scoring.

2) Is also unclear which predictors were used and passed statistical

significance (section Statistical analysis/ Data Synthesis Scoring).

3) In addition it would make the paper more interesting if individual examples

could be briefly highlighted in the discussion section. For example a treatment with

exceptional improvement and one example that does

not show a substantial clinical benefit across the

three chosen frameworks. This could allow to understand the

shortcoming of the frameworks. For example would it be possible to make

a venn diagram showing the overlap of studies with clinical benefit

and without between the frameworks (e.g. Figure 2a)? (Section Substantial clinical benefit)

4) One shortcoming seems to be also that the paper (e.g. the discussion section) does

not consider to discuss findings of the literature.

Currently only a single citation is given in the discussion section (e.g. include similar studies

of other countries, new measures/guidelines, statistical approaches,

electronic health records, follow-up etc.).

other comments

Background.

check sentence. There are different validated tools ... of a cancer drugs

Discussion.

We found that only around half of the included 'trials' showed ...

6. PLOS authors have the option to publish the peer review history of their article (what does this mean?). If published, this will include your full peer review and any attached files.

Reviewer #1: No

Reviewer #2: No

---

## [Author Response · Author response to Decision Letter 0]

8 Apr 2022

We have uploaded a document entitled "Response to Reviweres" where we address all point brough up during peer review and from the editor.

---

## [Editor Report · Decision Letter 1]

3 May 2022

Clinical Benefit of cancer drugs approved in Switzerland 2010 - 2019

PONE-D-21-28268R1

Dear Dr. Arnoud J Templeton,

We’re pleased to inform you that your manuscript has been judged scientifically suitable for publication and will be formally accepted for publication once it meets all outstanding technical requirements.

Kind regards,

Wen-Wei Sung, M.D., Ph.D.

Academic Editor

PLOS ONE

---

## [Editor Report · Acceptance letter]

10 May 2022

PONE-D-21-28268R1 

Clinical Benefit of cancer drugs approved in Switzerland 2010 - 2019 

Dear Dr. Templeton:

I'm pleased to inform you that your manuscript has been deemed suitable for publication in PLOS ONE. Congratulations! Your manuscript is now with our production department. 

Kind regards, 

on behalf of

Dr. Wen-Wei Sung 

Academic Editor

PLOS ONE